

# HPV-driven heterogeneity in cervical cancer: study on the role of epithelial cells and myofibroblasts in the tumor progression based on single-cell RNA sequencing analysis

Yunyun Zhang, Yu Zhang, Chenke Pan, Wenqian Wang and Yao Yu

Department of Ultrasound in Medicine, The Fourth Affiliated Hospital, Zhejiang University School of Medicine, Zhejiang University, Yiwu, Zhejiang, China

## ABSTRACT

**Background:** Cervical cancer (CC) is a neoplasia with a high heterogeneity. We aimed to explore the characteristics of tumor microenvironment (TME) for CC treatment.

**Methods:** HPV positive (+) and negative (−) samples from cervical cancer (CC) patients were sourced from the Gene Expression Omnibus (GEO) database. The single-cell RNA sequencing (scRNA-seq) data were processed and annotated for cell types utilizing the Seurat package. Following this, the expression levels and biological roles of the marker genes were analyzed applying real-time PCR (RT-PCR) and transwell assays. Furthermore, the enrichment of genes with significantly differential expressions and copy number variations was assessed by the ClusterProlifer and inferCNV software packages.

**Results:** Seven main cell clusters were classified based on a total of 12,431 cells. The HPV− CC samples exhibited a higher immune cell infiltration level, while epithelial cells and myofibroblasts had higher proportion in the HPV+ CC samples with extensive heterogeneity. Immune pathways including antigen treatment and presentation, immunoglobulin production and T cell mediated immunity were significantly activated in the HPV− CC group with lower cell cycle and proliferation activity. However, the anti-tumor immunity of these cells was inhibited in HPV+ CC group with higher cell proliferation activity. Moreover, the amplification and loss of CNVs also supported that these cells in HPV− CC samples were prone to anti-tumor activation. Further cell validation results showed that except GZMA, the levels of APOC1, CEACAM6, FOXP3, SFRP4 and TFF3 were all higher in CC cells Hela, and that silencing TFF3 could inhibit the migration and invasion of CC cells *in-vitro*.

**Conclusion:** This study highlighted the critical role of HPV infection in CC progression, providing a novel molecular basis for optimizing the current preventive screening and personalized treatment for the cancer.

Corresponding author
Yunyun Zhang,
zhangyunyun0517@zju.edu.cn

## INTRODUCTION

Cervical cancer (CC) is one of the most frequent gynecological carcinomas affecting the health of women worldwide, with about 570,000 new cases and 311,000 deaths each year (*Bray et al., 2018*). The CC incidence is associated with risk factors such regional factor (*Zheng et al., 2022*), tobacco smoking, promiscuity, unhygienic conditions and availability of early screening programs (*Moss, Liu & Feuer, 2017*). According to the least cancer statistics, new cases of CC and deaths are estimated to reach 13,820 and 4,360 in 2024, respectively (*Siegel, Giaquinto & Jemal, 2024*). At present, the incidence of CC has been reduced by the extensive use of Papanicolaou (Pap) screening and treatment of precursor lesions since the mid-1970s (*Siegel, Giaquinto & Jemal, 2024*), especially by the use of human papillomavirus (HPV) vaccine (*de Martel et al., 2017*). HPV infection has been identified as a major risk contributor to the occurrence and progression of CC as it is responsible for more than 95% of CC cases through sexually transmitted infection (STI) (*Adiga et al., 2021*). Clinically, the pathogenesis and metastasis of CC is dependent on the interaction between oncogenic HPV infection and host susceptibility (such as the presence of susceptibility loci variations) (*Bahrami et al., 2017*). Among 300 types of HPV genotypes, more than 70% of CC cases and 50% of high-grade cervical pre-cancer are related to the infection of high-risk HPV subtypes (HPV16 and 18) (*Tian et al., 2022*), while other CC progression and precancerous lesions are correlated with HPV types such as HPV 33, 31, 35, 39, 59, 58, 56, 68, 66, 52, 51, 45 (*Fani et al., 2020*).

In general, the pathogenesis of CC develops from the transmission of HPV to the basal layer of epithelial cells during sexual contact. Phenotypically, persistent oncogenic HPV infection will cause cervical intraepithelial lesions or cervical dysplasia formation and *in situ* carcinoma (*Wang, Huang & Zhang, 2018*), which gradually advances to neoplastic expansion and invasion cancer through the lymphatic and blood systems between 10 to 20 years after infection (*Martin & O'Leary, 2011*). Normally, around 70% of HPV-infected patients could recover spontaneously without showing lifestyle-threatening symptoms (*Kombe Kombe et al., 2020*). However, HPV employs several immune evasion mechanisms to promote carcinogenesis, for example, emergence of viral life cycle within epithelial cells and development of local immuno-suppression in viral proteins without viremia, cell death or inflammation (*Jain et al., 2023*). The oncogenic HPV is a type of double-stranded DNA, non-enveloped papillomaviridae virus, and its genome comprises three regions. Specifically, the early (E) region encode various precursor E proteins (E1, E2, E4, E5, E6, and E7) for early infection such as oncogenesis, viral DNA replication and cell cycle regulation (*Williams et al., 2011*), the late (L) region that encodes L1 and L2 proteins for viral capsid structure, and the long control region (LCR), which is a non-coding region for the regulation of viral replication and transcription (*Graham et al., 2006*). Under HPV infection, the virus genome will be integrated into the transcribed genomic region of host cells to significantly increase a series of viral products, particularly the E6 and E7 oncoproteins that contribute to the pathogenicity of virus through destroying the functions of retinoblastoma (Rb) and tumor suppressor proteins p53 (*Mahendra et al., 2022*) as well as the E1 and E2 proteins involved in the viral replication process (*Pal & Kundu, 2019*) and transcription regulation (*Pal & Kundu, 2019*).

Genotypic and phenotypic changes shape a unique tumor microenvironment (TME) during tumor progression (*Mendoza-Almanza et al., 2020*). Research indicated that lymphatic metastasis represents the primary form of CC metastasis, during which TME characterized by cancer-associated fibroblasts, tumor-associated macrophages, myeloid-derived suppressor cells, immune and inflammatory cells as well as blood and lymphatic vascular systems are all involved. These elements can facilitate the formation of lymphatic metastatic regions within immunosuppressive microenvironments or enhance lymphatic metastasis by promoting lymphangiogenesis and epithelial-mesenchymal transition (EMT) (*Wang et al., 2023*). Immunotherapy and anti-angiogenesis are used to treat advanced CC but the response rate remains poor (*Minion & Tewari, 2018*). To improve the current immunotherapy and systemic therapy strategies for CC, characteristics of TME and molecular profile of immune niche should be comprehensively analyzed. Single-cell RNA sequencing (scRNA-seq) allows an accurate assessment of gene expression profiles within thousands of distinct cells, facilitating the detection of potential variations in cell populations and providing new understanding on tumor diversity (*Zulibiya et al., 2023*). Thus, this study was conducted to investigate the characteristics of TME and a cluster of cells with specific immune niche associated with the progression of HPV+ and negative-CC. The scRNA-seq profiles of CC samples from the Gene Expression Omnibus (GEO) were analyzed to identify specific epithelial cells and myofibroblasts in the HPV+ CC samples and their significant pathways, copy number variations (CNVs) and heterogeneous cell profiles were also explored. Our study highlighted the critical role of HPV infection in tumor progression, revealing an underlying molecular mechanism for improving preventive screening of CC and further promoting the significance of HPV vaccination.

## MATERIALS AND METHODS

### Data acquisition

The scRNA-seq data including five adenocarcinoma of the cervix (ADC) samples in the GSE197461 cohort were obtained from the Gene Expression Omnibus (GEO, https://www.ncbi.nlm.nih.gov/) using the 10x Genomics and Illumina NovaSeq 6000 platforms for sequencing (*Qu et al., 2023*). Specifically, the ADC_1, ADC_2 and ADC_3 are human papillomavirus positive (HPV+) groups, while the ADC_4 and ADC_5 are the HPV− groups.

### Analysis of scRNA-seq landscape of the ADC samples

We performed the scRNA-seq analysis by using the Seurat R package (*Zulibiya et al., 2023*). The Read10X function was used to read the expression matrix of each sample, and the cells with 200~7,000 genes expressed in at least three cells and <10% proportion of mitochondrial gene were retained. Then, the SCTransform function was used to perform data normalization and the harmony R package was applied to remove the batch effects between samples after principal component analysis (PCA) dimension reduction. The RunUMAP function further was used for the Uniform Manifold Approximation and Projection (UMAP) dimension reduction and visualization. Cell clustering (setting

dims = 1:30, resolution = 0.1) was performed employing the FindNeighbors and FindClusters functions to subdivide epithelial cells and myofibroblasts at the dims = 1:20 and resolution = 0.2. Cell markers and annotation data were obtained from the CellMarker2.0 database.

## Gene set enrichment analysis

To elucidate the enrichment difference in the functions of epithelial cells and myofibroblasts in HPV+ and HPV− groups, the FindMarkers function of Seurat R package was used to identify significantly differential genes between the two groups based on significant $\log_2$ fold change value (*Reimand et al., 2019*). Subsequently, the gseGO function in the ClusterProlifer R package was used for the biological process (BP) analysis of gene ontology (GO) term and the gseKEGG function was used for the Kyoto Encyclopedia of Genes and Genomes (KEGG) enrichment analysis. Visualization of the pathways was achieved applying the GseaVis R package (*Reimand et al., 2019*).

## Analysis of differentially expressed genes

The differentially expressed genes (DEGs) analysis among each cell subgroups was performed by using the FindAllMarkers function (setting only.pos = TRUE, min. pct = 0.25, logfc.threshold = 0.25) (*Song et al., 2023*). After that, these DEGs were uploaded into the Database for Annotation, Visualization and Integrated Discovery database (DAVID, https://david.ncifcrf.gov/) for the enriched BPs (*Xiang et al., 2021*).

## Copy number variation analysis

The inferCNV R package was used for the copy number variation (CNV) analysis for epithelial cells and myofibroblasts in HPV+ and HPV− groups, in which the B cells in the healthy group served as a reference (setting cluster_by_groups = TRUE, analysis_mode = "subclusters", HMM_type = "i3", denoise = TRUE, HMM_report_by = "subcluster", HMM=TRUE) (*Lu et al., 2023*).

## Cell culture and intervention

The human normal cervical endometrial cell line HUCEC (CP-H059) and the CC cell line Hela (CL-0101) were both obtained from Procell Lifescience Co., Ltd. in Wuhan, China. Cells were cultured either in Roswell Park Memorial Institute-1640 medium (PM150110; Procell Lifescience Co., Ltd., Wuhan, China) or in Minimal Essential Medium (PM150410; Procell Lifescience Co., Ltd., Wuhan, China) at 37 °C with 5% $CO_2$. All the media were additionally supplemented with 10% fetal bovine serum (FBS, 164210; Procell Lifescience Co., Ltd., Wuhan, China) and 1% antibiotics (PB180120; Procell Lifescience Co., Ltd., Wuhan, China).

The small interfering RNAs for TFF3 (target sequence: 5′-AAGCAGAAAAAATACA TTTCAGG-3′) and the corresponding scramble negative control (5′-GGATAGCAGAA CAAAATTTCAAA-3′) were ordered from GenePharma (Shanghai, China) and transfected into Hela cells using Lipofectamine 2000 transfection kit (11668-027, Invitrogen, Carlsbad, CA, USA) following the manuals.

## Cell migration and invasion assays

A total of $1 \times 10^5$ Hela cells were plated in 6-well dishes in a culture medium containing 10% FBS to complete confluence. Subsequently, the cell monolayers were wounded with sterile 10 μL pipette tips, and the debris was eliminated by rinsing in PBS. Following this, the wounded monolayers were incubated in serum-free medium for 48 h (h). The wound healing rate was observed using an inverted optical microscope (DP27, Olympus Corp., Tokyo, Japan) and analyzed by ImageJ (version 1.48, National Institute of Health, Bethesda, MD, USA).

For the invasion assay, transwell chamber with 8 μm pore size culture inserts (3422; Corning, Inc., Corning, NY, USA) was used. In detail, a total of $1 \times 10^6$ cells/mL in 200 μL serum-free culture medium were placed in the upper chamber coated with Matrigel (354230, Corning, Inc., Corning, NY, USA), while the corresponding lower chamber was supplemented with 700 μL complete medium containing 10% serum. After 48 h, the cells invaded to the lower chamber were immobilized with 4% fixation solution (P1110; Solarbio, Beijing, China) and dyed with 0.1% crystalline violet staining (G1063, Solarbio, Beijing, China) for 20 min (min). The number of invaded cells at 48 h was calculated under the same optical microscope and processed with ImageJ.

## Real-time quantitative PCR

TRIzol reagent (15596-026; Invitrogen, Waltham, MA, USA) was applied to isolate and prepare total cellular RNA, which were reverse-transcribed into corresponding complementary DNA using a related assay kit (K16325; Solarbio, Beijing, China) according to the protocol. RT-qPCR was then performed using SYBR Green PCR Mastermix (SR1110; Solarbio, Beijing, China) at the following parameters: at 95 °C for 3 min, and 40 cycles at 95 °C for 20 s (s), at 60 °C for 30 s and at 72 °C for 60 s. GAPDH was an endogenous control, and the relative mRNA level was calculated with the $2^{-\Delta\Delta CT}$ method (*Livak & Schmittgen, 2001*). The primers (5′–3′) involved were listed below:

APOC1: Forward: AGGACAAGGCTCGGGAACTCAT; Reverse: GATGTCA CCCTTCAGGTCCTCA.

CEACAM6: Forward: GCCTCAATAGGACCACAGTCAC; Reverse: AGGGCTGCTA TATCAGAGCGAC.

FOXP3: Forward: GGCACAATGTCTCCTCCAGAGA; Reverse: CAGATGAA GCCTTGGTCAGTGC.

SFRP4: Forward: CTATGACCGTGGCGTGTGCATT; Reverse: GCTTAGGCGTTTA CAGTCAACATC.

TFF3: Forward: TCCAGCTCTGCTGAGGAGTACG; Reverse: ATCCTGGAGTCAAA GCAGCAGC.

GZMA: Forward: CCACACGCGAAGGTGACCTTAA; Reverse: CCTGCAA CTTGGCACATGGTTC.

GAPDH: Forward: GTCTCCTCTGACTTCAACAGCG; Reverse: ACCA CCCTGTTGCTGTAGCCAA.

## Statistical analysis

All calculations were performed using the R software (version 4.3.1) or GrpahPad Prism software (version 8.0.2). For bioinformatics analysis, the data between groups were compared with Wilcoxon's test, while student's $t$ test was applied for cell validation test. A $p < 0.05$ was considered as statistically significant.

# RESULTS

## Epithelial cell and myofibroblasts had high infiltration levels in HPV+ group

After quality control, normalization and UMAP dimensionality reduction, a total of seven main cell subgroups were obtained from 12,431 cells, including epithelial cells, myofibroblasts, plasma B cells, cytotoxic NK/T cells, regulatory T cells, B cells and myeloid cells (Fig. 1A). Cell clusters (Figs. S1A and S1B), gene numbers (Fig. S1C) and mitochondrial ratio (Fig. S1D) in different samples were visualized. The cluster-specific marker genes were shown in Figs. 1B and 1C, from which it could be observed that NKG7, KRT8, AGR2 and CLDN4 were significantly expressed in cytotoxic NK/T cells and epithelial cells. Immune cells such as regulatory T cells, cytotoxic NK/T cells, myeloid cells, plasma B cells and B cells had higher proportion in the HPV− group, while the epithelial cell and myofibroblasts had higher proportion in the HPV+ group (Figs. 1D and 1E). This indicated that the patients in the HPV− group with a high immune infiltration had a stronger immunity against cancer, while the immune ability in the HPV+ groups may be inhibited, and that the epithelial cells and myofibroblasts may play an important role in cervical cancer progression.

## Epithelial cells in the HPV+ group contributed to antigen presentation with a lower cell proliferation activity

Furthermore, we explored the differences of gene expression patterns of epithelial cells between the HPV+/− groups. The FindMarkers was used to identify significant genes, and GO enrichment analysis found that cytoplasmic translation, catecholamine secretion and transport process were enhanced in the HPV− group, while the HPV+ group has significantly enriched processes of nuclear division, chromosome division, positive regulation of cell cycle and detection of chemical stimulation (Fig. 2A) as well as remarkably upregulated HES1, SPHK1 and UBE2C in the positive regulation of cell cycle pathway (Fig. 2B). KEGG analysis revealed that the activities of ribosome, oxidative phosphorylation, antigen processing and presentation pathways in HPV− group were significantly enhanced, while the activity of cell cycle, cAMP signaling pathway, DNA replication, P53 pathway and steroid hormone pathway were noticeably inhibited (Fig. 2C). In HPV− group, the expressions of CDKN1A, SERPINB5 and TP53I3 in the P53 signaling pathway were downregulated (Fig. 2D). Activation of antigen-presenting pathways may be associated with immunotherapeutic response (Lee et al., 2020), and dysregulation of P53 signaling pathway is the focus of many targeted therapies (Liebl & Hofmann, 2021). To this end, we also observed that the marker genes of antigen processing and presentation pathway such as PSEM1, HSP90AB1 and HSPA1B were significantly

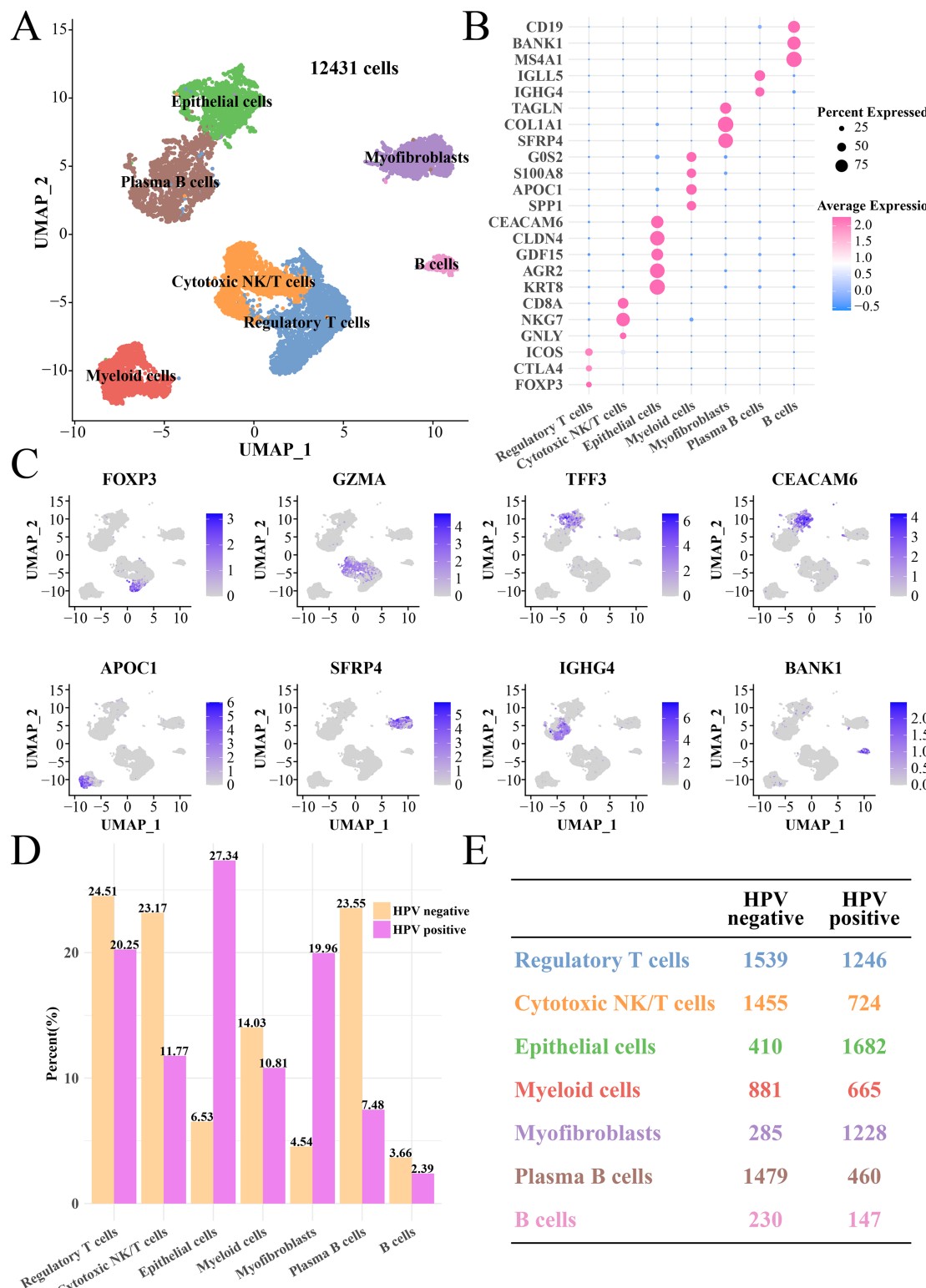

**Figure 1** **The single cell RNA-seq landscape of CC patients among HPV (+/−) groups.** (A) The UMAP plot of the seven mainly cell clustering. (B) The marker genes expression of specific cell clusters. (C) The tSNE plot of marker gene in each cell cluster. (D) The cell proportion analysis between HPV+ and HPV− groups. (E) The number of each cell type in the HPV+ and HPV− groups.

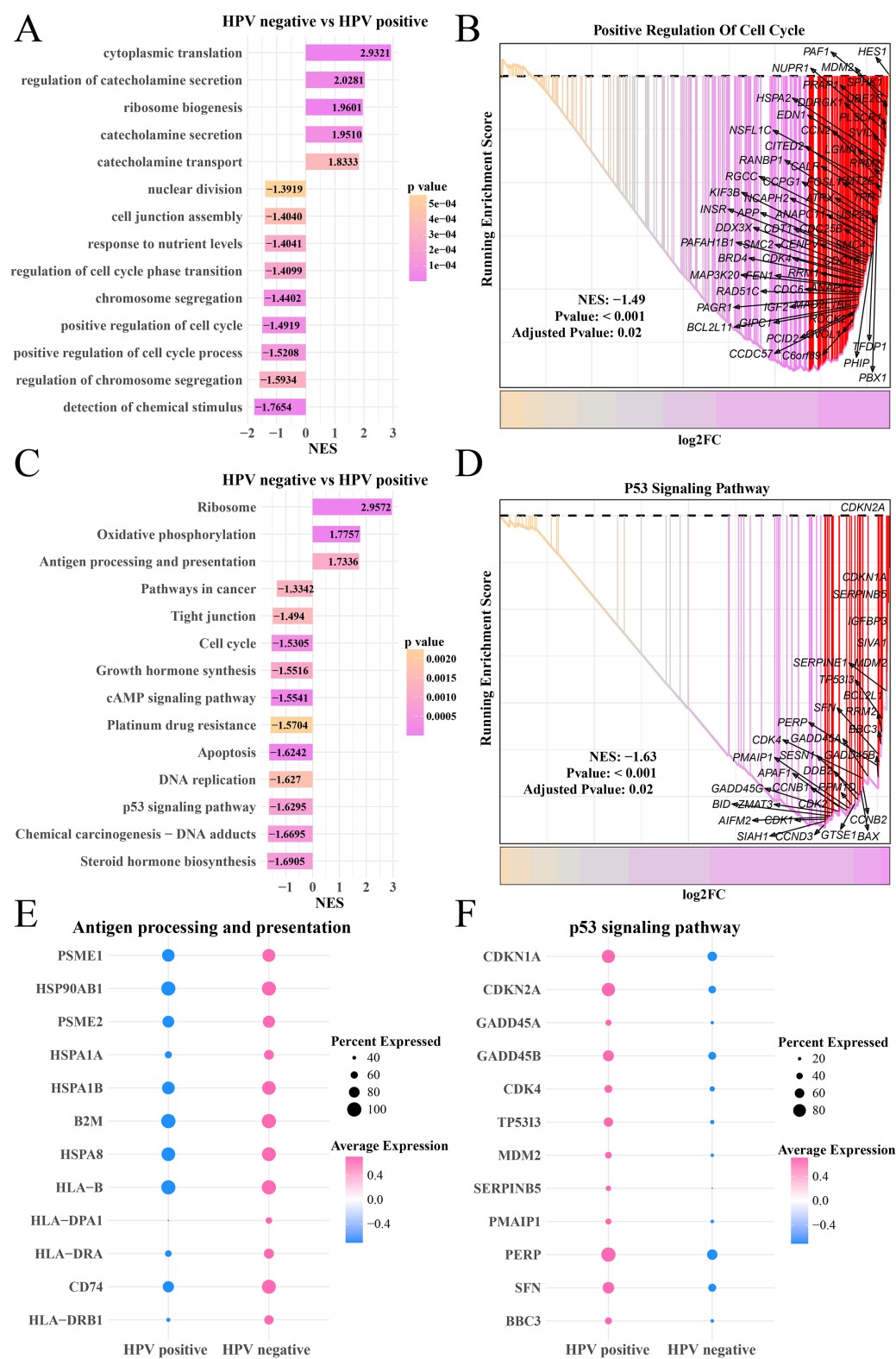

**Figure 2 The GSEA of epithelial cells in the HPV+ and HPV− groups.** (A) Biological process of epithelial cells in HPV− group compared with the HPV+ group. (B) Enrichment plot of the positive regulation of cell cycle pathway. (C) KEGG enrichment of epithelial cells in HPV− group compared with the HPV+ group. (D) Enrichment plot of the p53 signaling pathway. (E) Bubble plot of genes expression in the antigen processing and presentation pathway. (F) Bubble plot of genes expression in the p53 signaling pathway.

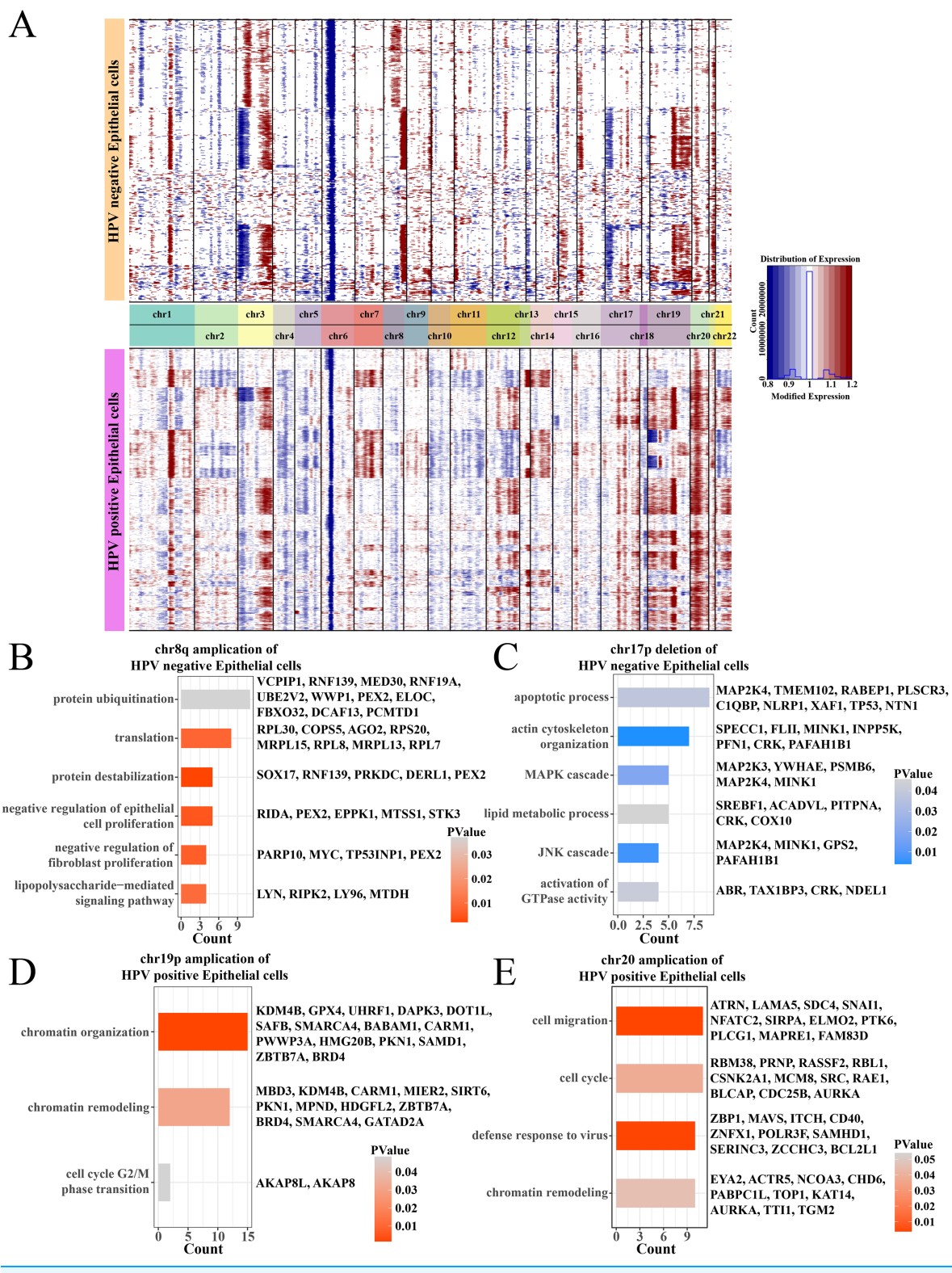

**Figure 3 The comparison of epithelial cell CNV in the HPV +/− groups.** (A) The heatmap of CNV change of epithelial cell CNV in the HPV+/− groups. (B) The enriched BP of chr8q amplification region in the HPV− group. (C) The enriched BP of chr17p deletion region in the HPV− group. (D) The enriched BP of chr19p amplification region in the HPV+ group. (E) The enriched BP of chr20 amplification region in the HPV+ group.

upregulated in HPV− group (Fig. 2E), while the marker genes of P53 signaling pathway such as CDKN (1A and 2A), GADD45B and PERP were significantly upregulated in the HPV+ group (Fig. 2F), suggesting that epithelial cells in the HPV− group were involved in antigen processing and presentation with a lower activity of cell proliferation, which resulted in the depletion of epithelial cells.

## Amplification of chr19p and chr20 may promote the epithelial cell proliferation in the HPV+ group

Analysis on the CNV of epithelial cells revealed amplifications in the chr1q and chr3q and deletions on the chr3p and chr6p between HPV+ and HPV− groups, and the difference was that the HPV− group had an amplification on the chr8q and a deletion on the chr17p (Fig. 3A). Specially, in the HPV− group, most negative regulation of epithelial cell proliferation genes such as RIDA, TP53INP1, STK3 and PEX2 were enriched on the chr8q amplification region (Fig. 3B), and the deletion region of chr17p contained genes associated with the MAPK, JNK cascade-related cell proliferation, for example, MAP2K3, PSMB6, GPS2 (Fig. 3C). These may help explain a low cell proliferation activity of epithelial cells in the HPV− group. In addition, the amplification region of chr19p and chr20 in the HPV+ group contained genes such as AURKA, BLCAP, CDC25B and MBD3, which were correlated with the chromatin remodeling, cell cycle, cell migration, and viral resistance response (Figs. 3D and 3E) that contributed the increase in the proportion of epithelial cells.

## Activated cell cycle and proliferation pathways enhanced the heterogeneity of epithelial cells in HPV+ group

The heterogeneity of epithelial cells in the HPV+ and HPV− groups were analyzed and the epithelial cells were further subdivided into three subgroups (epithelial cells 1, 2 and 3) (Fig. 4A) and their marker genes were showed in the (Fig. 4B). Interestingly, the epithelial cells 1 had higher proportion in the HPV− group, while the epithelial cells 2 and epithelial cells 3 had higher proportion in the HPV+ group (Fig. 4C). We analyzed the functions of significantly high-expressed genes in the three epithelial cell subgroups and found that cell cycle, cell migration, angiogenesis, viral resistance response and hypoxia pathways were significantly activated in the epithelial cells 1 ($p < 0.05$, Fig. 4D), while pathways such as oxidative stress, glycolysis and gluconeogenesis were mainly enriched in the epithelial cells 2 ($p < 0.05$, Fig. 4E) and insulin response, cold and heat response, progesterone response and corticosterone response pathways were enriched in the epithelial cells 3 ($p < 0.05$, Fig. 4F). These results indicated that the epithelial cells in the HPV+ group had higher cell proliferation and migration, resistance to viruses and angiogenesis activity. Correspondingly, enhanced oxidative stress and glycolysis, insulin, progesterone, and corticosterone activity all enhanced the tumor immunity in the HPV− group. This indicated that different epithelial cell subpopulations may play different roles in the progression and malignant transformation of CC by participating in specific biological processes.

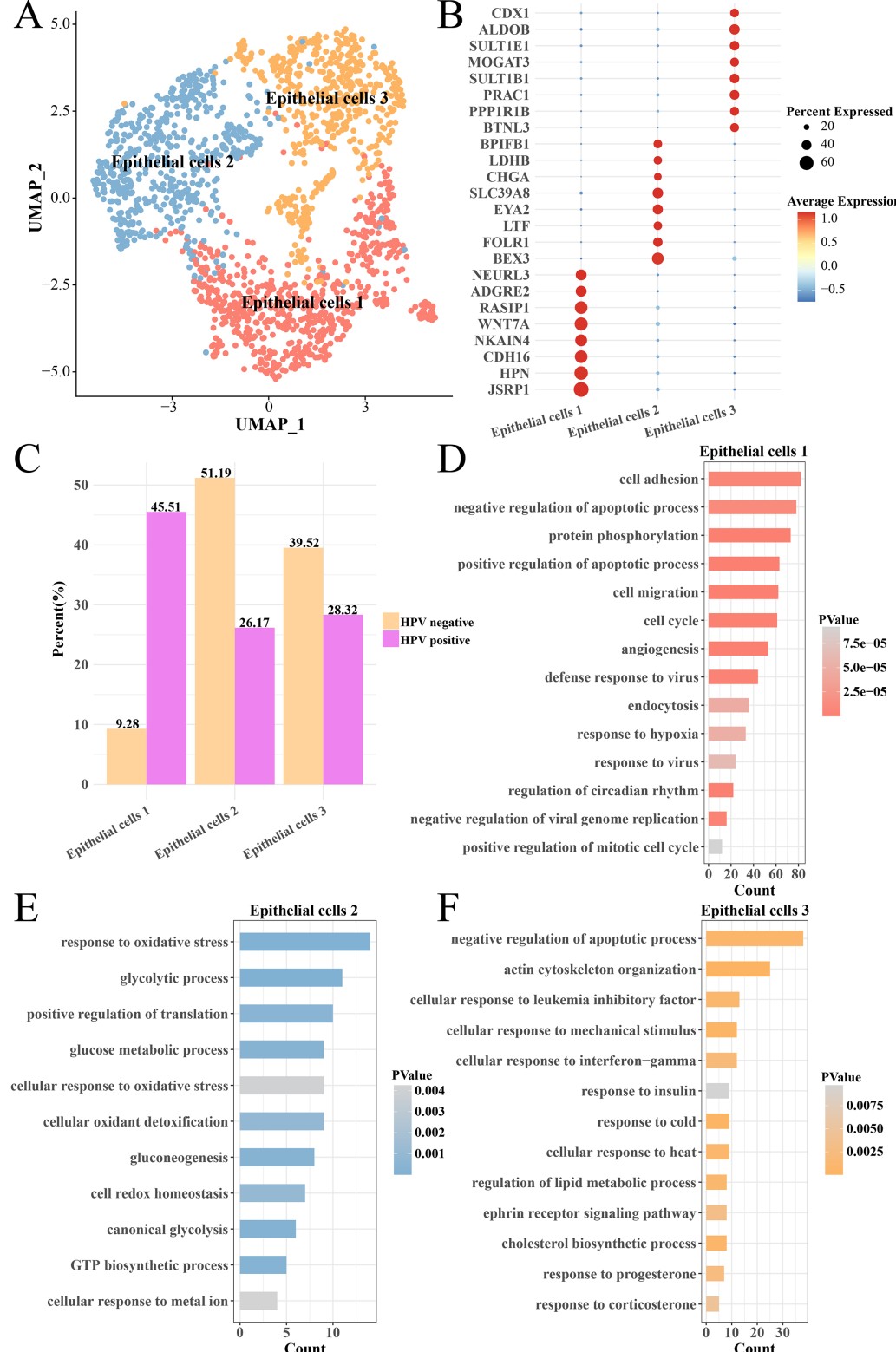

**Figure 4 Heterogeneity analysis of epithelial cells in HPV+/− groups.** (A) UMAP plot of epithelial cell sub-groups. (B) Bubble map of gene expression levels of epithelial cell sub-groups. (C) Proportion of epithelial cell sub-group between HPV + and HPV− groups. (D) The enriched BP of the significantly expressed genes in epithelial cells 1. (E) The enriched BP of the significantly expressed genes in epithelial cells 2. (F) The enriched BP of the significantly expression genes in epithelial cells 3.

## Activated tumor immunity increased the depletion of myofibroblasts in the HPV− group

Analysis on the gene expression pattern in the myofibroblasts in the HPV+ and HPV− groups identified significantly expressed genes. The GO enrichment analysis showed that in the HPV− group, immunoglobulin production, T- and B-cell-mediated immunity, antigen processing and presentation pathways were mainly activated, but cAMP cellular response, epidermal morphogenesis and adaptability of striated muscle and skeletal muscle pathways were significantly inhibited (Fig. 5A), HLA-F, DRB1, CTSH and HLA-C, A and E involved in T cell-mediated cytotoxicity were upregulated ($p < 0.05$, Fig. 5B). KEGG enrichment analysis showed that pathways including antigen processing and presentation, autoimmune thyroid disease and systemic lupus erythematosus, human papillomavirus infection were significantly activated in the HPV− group, while steroid hormone production, chemical cancer generation pathways were remarkably suppressed ($p < 0.05$, Fig. 5C). The genes (CYP1B1, EPHX1, GSTM5 and PTGS2) involved in the chemical carcinogenesis-DNA adducts in were noticeably upregulated the HPV+ group ($p < 0.05$, Fig. 5D). The bubble plot showed that the genes of the antigen processing and presentation (HLA-A, B, C and E) were significantly expressed in the HPV− group (Fig. 5E) and the genes of chemical carcinogenesis-DNA adducts (GSTM3 and 2, EPHX1) were significantly expressed in the HPV+ group (Fig. 5F), indicating that tumor immune activation may accelerate the depletion of myofibroblasts in the patients in the HPV− group.

## Activation of anti-cancer immunity in myofibroblasts 1 in the HPV− group

Analysis on the CNV of myofibroblasts revealed an extensive deletion of on the chr6p in the two HPV groups, indicating that there was no obvious difference of CNV in myofibroblasts between HPV+ and HPV− groups. The myofibroblasts were further subdivided into the subgroups of myofibroblasts 1 and 2 (Fig. 6A), with the HPV+ group having a higher proportion of myofibroblasts 2 (Fig. 6B). Pathway enrichment analysis revealed that cell migration, positive regulation of interleukin-8 production, immune system process and positive regulation of T cell-mediated cytotoxicity were remarkably enriched in the myofibroblasts 1 (Fig. 6C), while signal transduction, cell adhesion, cell growth and proliferation, angiogenesis pathways were mainly enriched in the myofibroblasts 2 (Fig. 6D). The genes (HLA-E, A and B, B2M) involved in positive regulation of T cell-mediated cytotoxicity pathway were significantly expressed in the myofibroblasts 1 ($p < 0.05$, Fig. 6E), while the marker genes of positive regulation of cell division pathway including the PGF, PTN, MDK and PGF7 were significantly expressed in the myofibroblasts 2 ($p < 0.05$, Fig. 6F). These results suggested that the activated cell growth and proliferation, angiogenesis pathways could increase the number of myofibroblasts 2 in the HPV+ group, while activated immune response pathway of myofibroblasts 1 contributed to the anti-cancer immunity and promoted the depletion of myofibroblasts.
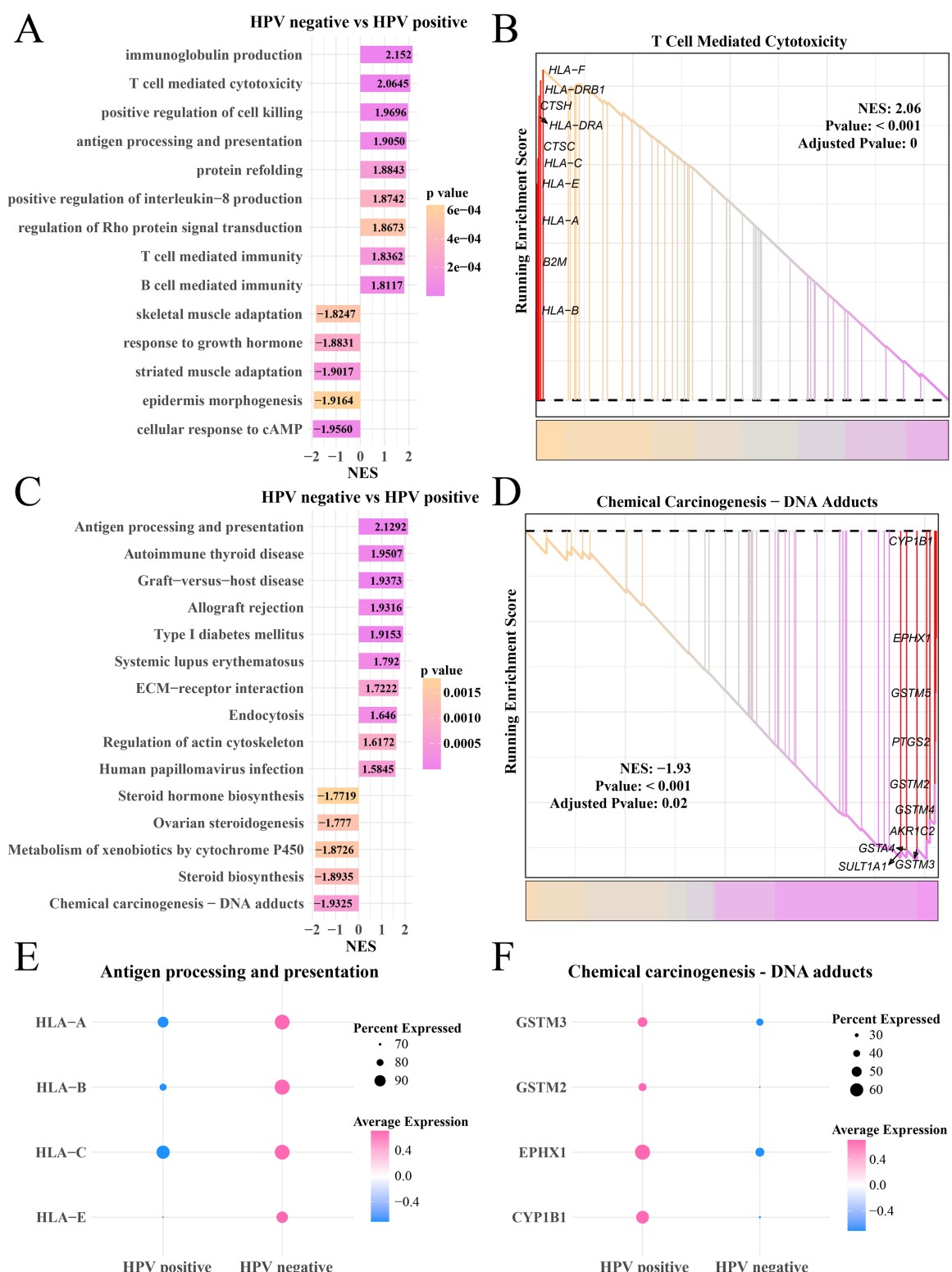

**Figure 5 GSEA analysis of myofibroblasts between HPV+/− groups.** (A) Biological process of myofibroblasts in HPV− group compared with the HPV+ group. (B) Enrichment plot of the T cell mediated cytotoxicity pathway. (C) KEGG enrichment of myofibroblasts in HPV− group compared with the HPV+ group. (D) Enrichment plot of the chemical carcinogenesis pathway. (E) Bubble plot of genes expression in the antigen processing and presentation pathway. (F) Bubble plot of genes expression in the chemical carcinogenesis-DNA adducts pathway.

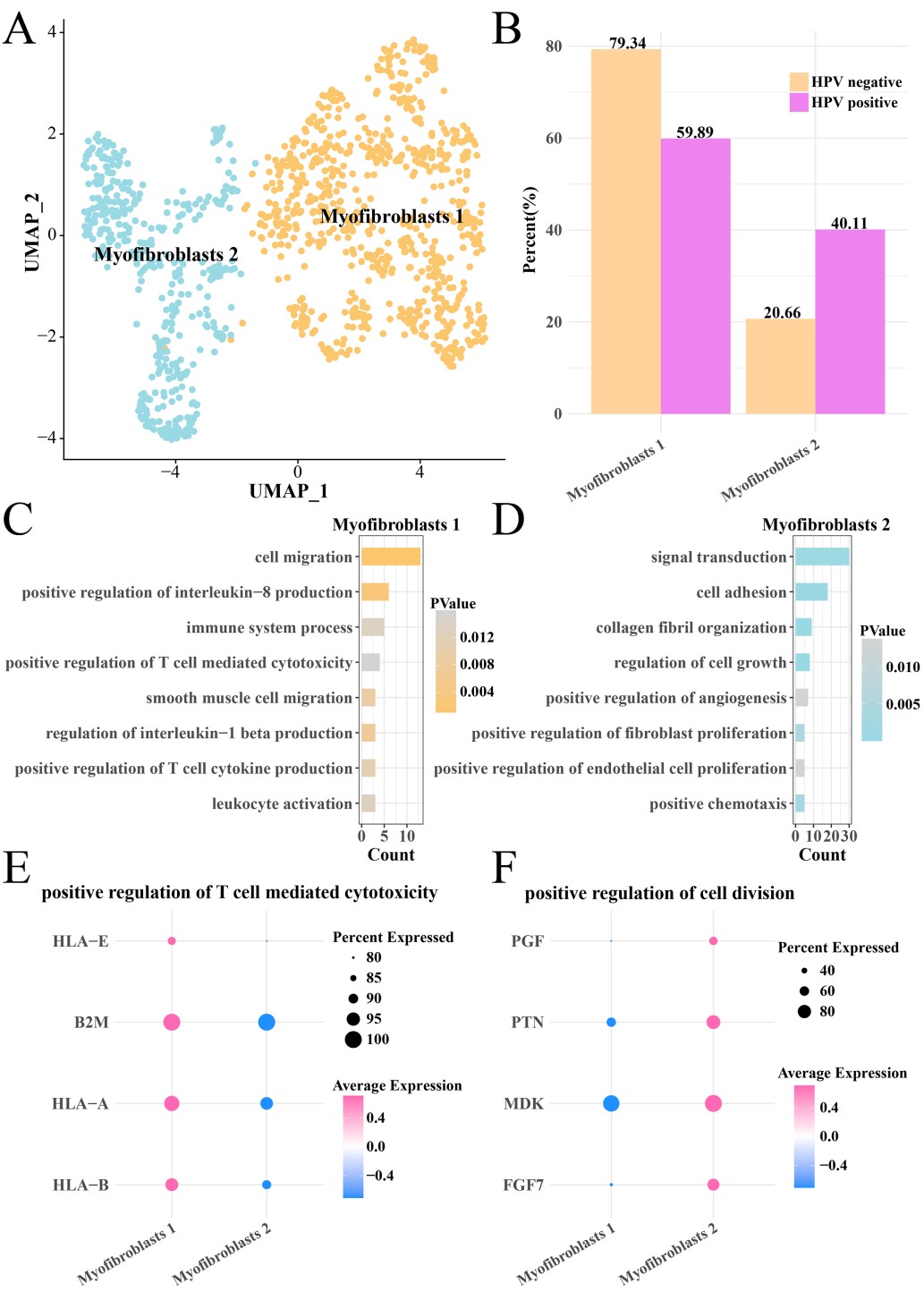

**Figure 6 Heterogeneity analysis of myofibroblasts in HPV+/− groups.** (A) UMAP plot of myofibroblasts sub-groups. (B) Proportion of myofibroblasts sub-group between HPV+ and HPV− groups. (C) The enriched BP of the significantly expressed genes in myofibroblasts 1. (D) The enriched BP of the significantly expressed genes in myofibroblasts 2. (E) Bubble plot of genes expression in the positive regulation of T cell-mediated cytotoxicity pathway. (F) Bubble plot of gene expression in the positive regulation of cell division.

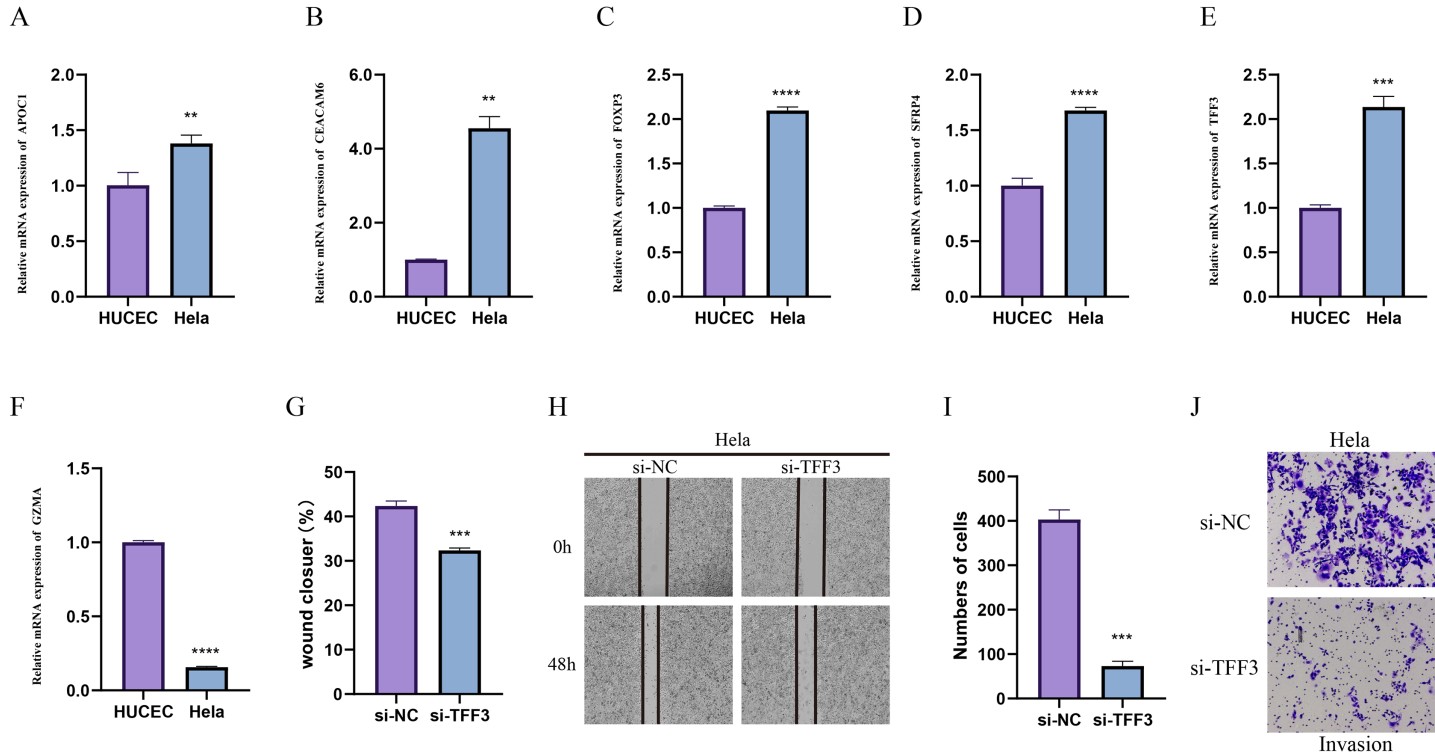

**Figure 7 *In-vitro* cell validation.** (A–F) Relative mRNA levels of APOC1 (A), CEACAM6 (B), FOXP3 (C), SFPR4 (D), TFF3 (E) and GZMA (F) in normal cervical endometrial cell line HUCEC and CC cell line Hela. (G–I) Effects of TFF3 silencing on the *in-vitro* migration and invasion of CC cell line Hela. **$p < 0.01$, ***$p < 0.001$, ****$p < 0.0001$.

### *In-vitro* cell validation

Due to the importance of method validation in biomarker analysis (*Seyfinejad & Jouyban, 2022*), the reliability of our identified marker genes based on HPV+ and HPV- samples were validated. To further validate the reliability of our identified marker genes based on HPV+ and HPV− samples, we assessed the levels of markers for specific cellular subpopulations including APOC1, CEACAM6, FOXP3, SFRP4, TFF3, and GZMA in CC cell line Hela and normal cervical endometrial cell line HUCEC. As shown in Figs. 7A–7F, we observed that except GZMA, the mRNA levels of all the rest markers were expressed at higher levels in Hela cells than in HUCEC cells ($p < 0.01$). Also, the results from wound healing assay (Figs. 7G and 7H) and Transwell assay (Figs. 7I and 7J) showed that the silencing of TFF3 visibly suppressed the migration and invasion of Hela cells *in vitro* ($p < 0.001$; Figs. 7G–7J).

## DISCUSSION

CC is pathogenetically more complex than other cancers largely due to its association with HPV (*Sundaram et al., 2021*). Recently, researchers have demonstrated that the upregulation of miR-200a-3p suppresses the cell growth, migration, and invasion of HPV− C33A cells but it enhances the growth and metastasis of HPV+ Siha and Hela cells. This suggested that miR-200a-3p plays a distinct dual regulatory role in HPV− compared to HPV-positive cervical cancer cells (*Chen et al., 2022*). In addition, *Stuqui et al. (2016)*

indicated that HTRA1 overexpression inhibits the growth of cells in the HPV− line but promotes cell growth of the HPV+ line. These findings demonstrated that analysis on the gene differences between HPV− and HPV+ could help understand the underlying molecular mechanisms and develop targeted therapies for HPV-infected CC. Therefore, based on the HPV+ and HPV− CC samples from public databases, this study applied scRNA-seq to reveal the heterogeneity of TME in CC patients to explore the pathogenesis of CC from the aspect of HPV infection.

TME is a complex immune microecology composed of various immune cells that produce anti-tumor response or immunosuppressive effect in tumors (*Shimizu et al., 2018*). Studies showed that cancer-associated fibroblasts (CAFs) are the primary contributor to the phenotypic heterogeneity in CC including in the EMT process, for instance, activated normal fibroblasts, and EMT is known to enhance the invasion, proliferation and metastasis of tumor (*Wen et al., 2023*). However, at present, we lacked a comprehensive analysis on the heterogeneity between the HPV+ and HPV− groups in clinical practice (*The Cancer Genome Atlas Research Network, 2017*). Under the cancer staging system of International Federation of Gynecology and Obstetrics (FIGO), around 15–38% of HPV− patients have advanced phenotype and significantly poor prognosis with highly lymphatic invasion than the HPV+ patients. The molecular etiology of HPV− is unknown but the prominent role of TP53, PIK3CA and CDKN2A mutation in HPV− CC cases has been emphasized in several studies (*Nicolás et al., 2019*; *Li et al., 2017*). Higher mortality and more advanced progression of HPV-CC cases indicated that these patients might have distinct pathways that differ from widely characterized HPV-dependent pathways (*Nicolás et al., 2019*). Compared to the HPV+ CC cases, our study found that the HPV− CC group had higher immune infiltration of cytotoxic NK/T cells and plasma B cells, which were associated with the anti-tumor response. This indicated that most chemicals (chemokines and cytokines) for immune cell recruitment were secreted in the HPV− CC cases (*Pfaffenzeller, Franciosi & Cardoso, 2020*), which may be a unique mechanism that accelerated the depletion of anti-tumor immune cells in the HPV+ CC cases.

The epithelial cells and myofibroblasts accounted for greater proportions in the HPV+ CC cases and exhibited a high heterogeneity. CC usually originates from the epithelial cells of the cervix and is driven by HPV infection. The malignant transformation of epithelial cells will lead to uncontrolled cell proliferation and tumor formation (*Wang et al., 2023*). In addition, myofibroblasts not only play an important role in wound healing process in normal tissues, but also causes chronic inflammatory responses that could promote cancer progression (*Sferra et al., 2018*). EMT is a process during which epithelial cells lose their normal characteristics and transform into mesenchymal cells to promote the invasion and metastasis of tumors (*Qureshi, Arora & Rizvi, 2015*), indicating that these two cells could enhance the potential of invasion and metastasis in the HPV+ CC. The functional enrichment analysis showed that nuclear division, cell cycle regulation phase transition, p53 signaling pathway and chromosome segregation activity of epithelial cells were enhanced in the HPV+ CC. These results were consistent with the etiological role of host integration of HPV. The functions of E2 proteins are disrupted by the integration of DNA

to cause loss of control of E6 and E7 proteins and promote aberrant viral gene expression (*Gupta & Mania-Pramanik, 2019*). Study reported that the E2 proteins are associated with the mitochondria crest proteins such as cytochrome (Cyt) and ATP synthase and oxidative stress (*Cruz-Gregorio et al., 2018*), which may be a contributor to CC pathogenesis due to its role in DNA damage to increase HPV genome integration (*Jelic et al., 2021*). E6 oncoprotein has a neoplastic effect *via* promoting the ubiquitination of p53 degradation during the apoptosis of HPV-infected cells (*Celegato et al., 2021*), a process that involves the evasion of cell cycle checkpoints for tumor proliferation (*Patel et al., 1999*). This study found that cytoplasmic translation, ribosome, oxidative phosphorylation and antigen procession and presentation activity of epithelial cells and myofibroblasts were enhanced in the HPV− CC cases, indicating that these cells may be involved in the production of cytokines and chemokines and the recruitment of other immune cells for anti-tumor response. Moreover, the CNV and heterogeneity analyses showed the amplification in the HPV− CC cases inhibited the expressions of genes related to cell proliferation (RIDA, TP53INP1, STK3 and PEX2) and those of the genes (AURKA, BLCAP, CDC25B and MBD3) related to chromatin remodeling, cell cycle regulation and migration, and antiviral reaction in the HPV+ CC. The epithelial cells and myofibroblast with a lower activity of cell proliferation enhanced the anti-tumor response in the HPV− group, while the HPV+ group was prone to cell proliferation activation for the tumor invasion and metastasis. This indicated that the development of targeted therapies or drugs against these genes or related signaling pathways may be an effective strategy for inhibiting the progression of CC.

## CONCLUSION

In summary, the present study provided novel insights into the heterogeneity of different cellular subpopulations in CC applying scRNA-seq analysis, particularly the functional differences between epithelial cells and myofibroblasts in HPV+ and HPV− samples. We found that HPV infection affected cell proliferation, migration and immune escape mechanisms in CC, and also observed changes in the expressions of key genes and signaling pathways in the TME that drove tumor progression and malignant transformation. The study improved the current understanding of the oncogenic role of HPV in CC, offering new insights for the development of targeted therapies for the cancer.

## ABBREVIATIONS

| | |
|---|---|
| **CC** | Cervical cancer |
| **scRNA-seq** | single cell RNA-seq |
| **HPV** | human papillomavirus |
| **FDA** | Food and Drug Administration |
| **STI** | sexually transmitted infection |
| **TME** | tumor microenvironment |
| **GEO** | Gene Expression Omnibus |
| **HPV (+)** | HPV positive |
| **HPV (−)** | HPV negitave |
| **CNV** | copy number variation |

| ADC | adenocarcinoma of the cervix |
| PCA | Principal Component Analysis |
| UMAP | Uniform Manifold Approximation and Projection |
| GSEA | Gene Set Enrichment Analysis |
| BP | biological process |
| GO | gene ontology |
| KEGG | Kyoto Encyclopedia of Genes and Genomes |
| DEGs | differentially expressed genes |
| DAVID | Database for Annotation, Visualization and Integrated Discovery database |
| WHO | World Health Organization |
| CAFs | cancer-associated fibroblasts (CAFs) |
| EMT | epithelial-to-mesenchymal transition (EMT) |
| FIGO | Federation of Gynecology and Obstetrics (FIGO) |

### Funding
This work was supported by foundation of Zhejiang Provincial Education Department (No. Y202248779). The funders had no role in study design, data collection and analysis, decision to publish, or preparation of the manuscript.

### Grant Disclosures
The following grant information was disclosed by the authors:
Zhejiang Provincial Education Department: Y202248779.

### Competing Interests
The authors declare that they have no competing interests.

### Author Contributions
- Yunyun Zhang conceived and designed the experiments, performed the experiments, analyzed the data, prepared figures and/or tables, authored or reviewed drafts of the article, and approved the final draft.
- Yu Zhang performed the experiments, prepared figures and/or tables, and approved the final draft.
- Chenke Pan conceived and designed the experiments, analyzed the data, prepared figures and/or tables, authored or reviewed drafts of the article, and approved the final draft.
- Wenqian Wang performed the experiments, authored or reviewed drafts of the article, and approved the final draft.
- Yao Yu conceived and designed the experiments, performed the experiments, analyzed the data, authored or reviewed drafts of the article, and approved the final draft.

### Data Availability
The datasets generated and/or analyzed during the current study are available at GEO: GSE197461.

The raw data is available at GitHub and Zenodo: - https://github.com/123zhang-dev/Raw-data.git.

- 123zhang-dev. (2024). 123zhang-dev/Raw-data: Raw data (v.1.1.0). Zenodo. https://doi.org/10.5281/zenodo.12602716.

## Supplemental Information

Supplemental information for this article can be found online at http://dx.doi.org/10.7717/peerj.18158#supplemental-information.

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
