# Peer review of "HPV-driven heterogeneity in cervical cancer: study on the role of epithelial cells and myofibroblasts in the tumor progression based on single-cell RNA sequencing analysis"

_PeerJ, doi:10.7717/peerj.18158_

## Round 0.1 · original submission · Major Revisions

1. Please discuss the research significance and implications of your findings, particularly in terms of contributing to the development of cervical cancer treatment strategies and understanding the roles of epithelial cells and myofibroblasts in cervical cancer progression.

2. Provide a more systematic description of the analytical tools, experimental procedures, conditions, materials, and reagents used in the study.
3. Please enhance the introduction by elaborating on the rationale for selecting cervical cancer for single-cell clustering, the significance of conducting single-cell transcriptome analyses in cervical cancer research, and the mechanism of intercommunication between cell subpopulations in the cervical cancer tumor microenvironment.

4. Address the potential impact of sample size differences between the HPV (+) and HPV (-) groups on the results, particularly in Figure 1B.
5. Please justify the focus on specific pathways in Figure 2, such as antigen presentation and p53 signaling, despite the identification of other significantly different pathways.

6. Please clarify the relationship between cell proliferation differences and HPV (+) and HPV (-) infection, discussing any previous reports on this aspect.
7. Please clearly state the gene markers of cell subgroups used in the in vitro cell validation, their relationship to specific cell clusters, and the implications of performing in vitro validation.

8. Provide a summary statement of the functional enrichment analyses in Figure 4 and formulate reasonable hypotheses.

9. Discuss how your study contributes to preventive screening and HPV vaccination efforts for cervical cancer, highlighting the importance of your work in the introduction and discussion sections.

**Language Note:** The review process has identified that the English language must be improved. PeerJ can provide language editing services - please contact us at [email protected] for pricing (be sure to provide your manuscript number and title). Alternatively, you should make your own arrangements to improve the language quality and provide details in your response letter. – PeerJ Staff

Reviewer 1 ·

Basic reporting

In this study, the author explores the potential link between the epithelial cells and myofibroblasts in the development of cervical cancer. Risk factors and signatures in cervical cancer were identified through bioinformatics methods. The experimental design is rigorous. However, there are still some deficiencies in details in the manuscript.
1. How does this paper promote the development of CC treatment strategies.
2. What are the advantages and disadvantages of this article.
3. In the conclusion of abstract section, what is the important research significance of this paper.
4. Epithelial cells and myofibroblast were the 2 types of important cells identified, what are their basic cellular functions and characteristics, please describe it appropriately in the discussion.

Experimental design

5. Which factors may affect the antigen presenting ability of Epithelial cells in the HPV (+) group, what is their status in normal cells.
6. In Figure 1B, is there any influence on the results of this paper due to the large difference in the number of samples, samples HPV (-) was obviously more than sample HPV (+). Please explain it.
7. In the Figure 2, the differential gene expression pattern analysis identified several significantly different pathways, such as catecholamine transport, why only the antigen presentation and p53 signaling pathways are important.
8. Is cell proliferation difference the main feature of HPV (+) and HPV (-) infection. Has it been reported in previous articles.

Validity of the findings

9. In the vitro cell validation, the gene markers of cell subgroups do not seem to have specifically appeared in the article, please add clearly in the article.
10. The relationship of these marker genes to specific cell clusters is not clear, What are the implications of performing in vitro validation.

Reviewer 2 ·

Basic reporting

no comment

Experimental design

no comment

Validity of the findings

no comment

Additional comments

This study aimed to elucidate the potential cell types affecting cervical cancer (CC) progression by single-cell transcriptome analysis and to reveal the regulatory mechanisms of these cell types in cancer progression. This study first acquired single-cell transcriptome data of CC from public databases and analyzed the clustering of cell subpopulations. Subsequently, potential cell subpopulations affecting CC were acquired and these cell types were analyzed by CNV analysis and functional enrichment. Finally, the joint experiments revealed the tangible regulatory roles of these cellular subpopulation marker genes in the malignant progression of cancer. In conclusion, this is a bioinformatics joint experimental study, but the following issues still need to be addressed before publication:
1. This study did not include a very systematic description of the various analytical tools and experimental procedures used in this study in the abstract section, and it is recommended that this be added to make the text more complete.
2. Why was CC selected for single-cell clustering in this study? There are many well-established studies on the heterogeneity of CC, won't clustering them in this paper be too homogeneous? Please provide an explanatory note on this.
3. The experimental section of this study suggests that the experimental methodology be supplemented with a complete description of the experimental conditions, experimental materials, and reagents used in this experiment, respectively. In addition, it is recommended that the experimental description be adjusted so that it does not duplicate the methods of existing studies.
4. It is recommended that the introduction section be supplemented with information about the significance of carrying out single-cell transcriptome analyses and a refinement of the significance of single-cell transcriptome analyses in CC research, so as to make the intention of the whole text clearer.
5. Much of the introductory section of this paper describes the role of HPV in CC, but the title of this paper does not seem to address this aspect, and it is recommended that the title of this paper be revised.
6. The introductory section of this study states that the pathogenesis of CC begins with the spread of HPV to basal epithelial cells during skin contact activity, so what is the mechanism of intercommunication between cell subpopulations in the tumor microenvironment of CC? What kind of mechanism leads to this outcome? Please elaborate clearly in the original article.
7. The description of the results in Figure 4 appears to be too general, and the conclusion does not elucidate what regulatory function is associated with the heterogeneity of epithelial cells; thus, it is recommended that a summary statement be added to the results of the functional enrichment analyses and that reasonable hypotheses be formulated.
8. Currently, preventive screening for CC and HPV vaccination are more commonly publicized, and does the conduct of the study in this article contribute to this aspect? Please highlight the importance of the development of this article by focusing on the introduction and discussion sections.
9. The conclusion section of this paper only briefly summarizes the contents of the full paper, which does not appear to be in-depth enough, and it is recommended that a follow-up polarization on this study be added to this section, including what experiments were conducted to validate the important conclusions of this paper.
10. Immune cells in the tumor microenvironment play an important role in regulating tumor progression, and the heterogeneity of the tumor microenvironment is also one of the difficulties in the treatment of CC, which is described in the discussion section of this paper is a wise move. However, the description of the tumor microenvironment in the Discussion section of this paper is too general, and it is recommended that more literature be added to illustrate which typical cellular interactions are effectively regulated for CC progression.

---

## Round 0.2 · accepted · Accept

All comments have been fully addressed by authors, and I think this paper can be accepted for publication.

Reviewer 1 ·

Basic reporting

no comment

Experimental design

no comment

Validity of the findings

no comment

Additional comments

In this study, the authors explored the potential link between epithelial cells and myofibroblasts in the development of cervical cancer. The risk factors and characteristics of cancer were identified through bioinformatics methods. And experimental verification was conducted. Overall, the experimental design was rigorous, the writing was fluent, and it was in line with readers' habits. The results and conclusions were rigorous, and the discussion was reasonably extended. In the revised version, the author provided detailed responses to the reviewer's comments, and I do not have any new comments.

Reviewer 2 ·

Basic reporting

no comment

Experimental design

no comment

Validity of the findings

no comment

Additional comments

Cervical tumors are the fourth most common cancer in women worldwide, accounting for nearly 8% of all cancer deaths in women each year. Most cervical cancers are caused by human papillomavirus (HPV); The characteristics of HPV negative cervical cancer are: a differentiated molecular profile with low proliferative activity, p53 immunostaining, decreased expression of cyclin dependent kinase inhibitors (such as p16, p14, and p27), and changes in PTEN, p53, KRAS, CTNNB1, ARID1A, and ARIDSB. HPV negative cervical cancer is associated with adenocarcinoma and squamous histological subtypes, with early lymph node involvement, further metastasis, and generally poorer oncological outcomes. So far, it is difficult to explain the development of HPV negative cervical cancer, but the "virus hit and run theory" can explain the loss of viral genomes in these cases. The "hit and run theory of viruses" proposes that once viral infection causes sufficient cellular changes, the expression of viral proteins or viral infection of tumors no longer needs to continue, and therefore the virus may be lost during cancer progression. E6/E7 oncogenes begin the carcinogenic process, but as mutations accumulate over time, transcription of viral genes is no longer necessary, resulting in their loss. In addition, some people have proposed that the "virus hit and run theory" of tumor occurrence may also leave permanent traces through epigenetic disorders. Chromatin remodeling may expose viral hotspots, thereby impairing transcriptional regulation, DNA repair, and permanent epigenetic changes in infected cells. In this study, single-cell transcriptome analysis was used to elucidate the potential cell types that affect the progression of HPV negative and positive cervical cancer, and to reveal the regulatory mechanisms of these cell types in cancer progression. Combined with experimental verification, the actual regulatory role of these cell subpopulation marker genes in cancer malignancy progression was revealed. The manuscript has a certain degree of innovation and significance in exploring the molecular mechanisms of cervical cancer, and overall meets the publishing requirements.